# Impact of Sapphire Step Height on the Growth of Monolayer Molybdenum Disulfide

**DOI:** 10.3390/nano13233056

**Published:** 2023-11-30

**Authors:** Jie Lu, Miaomiao Zheng, Jinxin Liu, Yufeng Zhang, Xueao Zhang, Weiwei Cai

**Affiliations:** 1College of Physical Science and Technology, Xiamen University, Xiamen 361005, China; 19820190154689@stu.xmu.edu.cn (J.L.); zhengmiaomiao@stu.xmu.edu.cn (M.Z.); jxliu@stu.xmu.edu.cn (J.L.); yufengzhang@xmu.edu.cn (Y.Z.); 2Jiujiang Research Institute, Xiamen University, Jiujiang 360404, China

**Keywords:** chemical vapor deposition, monolayer MoS_2_, morphology, sapphire step height

## Abstract

Although the synthesis of molybdenum disulfide (MoS_2_) on sapphire has made a lot of progress, how the substrate surface affects the growth still needs to be further studied. Herein, the impact of the sapphire step height on the growth of monolayer MoS_2_ through chemical vapor deposition (CVD) is studied. The results show that MoS_2_ exhibits a highly oriented triangular grain on a low-step (0.44–1.54 nm) substrate but nanoribbons with a consistent orientation on a high-step (1.98–3.30 nm) substrate. Triangular grains exhibit cross-step growth, with one edge parallel to the step edge, while nanoribbons do not cross steps and possess the same orientation as the step. Scanning electron microscopy (SEM) reveals that nanoribbons are formed by splicing multiple grains, and the consistency of the orientation of these grains is demonstrated with a transmission electron microscope (TEM) and second-harmonic generation (SHG). Furthermore, our CP2K calculations, conducted using the generalized gradient approximation and the Perdew–Burke–Ernzerhof (PBE) functional with D3 (BJ) correction, show that MoS_2_ domains prefer to nucleate at higher steps, while climbing across a higher step is more difficult. This work not only sheds light on the growth mechanism of monolayer MoS_2_ but also promotes its applications in electrical, optical, and energy-related devices.

## 1. Introduction

Molybdenum disulfide (MoS_2_) has garnered great attention due to its immense potential in various applications, such as transistors, photodetectors, and catalysis [1,2,3,4]. The preparation of MoS_2_ is crucial for achieving these applications. The simulation works reveal that the role of substrates, capping layers, etc., is significant to the growth of MoS_2_ and related materials, which guides experimental works [5,6,7]. Currently, two types of substrates are generally used in synthesis: amorphous and single-crystal. The former is represented by silica and glass [8,9]. The crystal orientation is disordered when growing on these substrates, which results in polycrystalline films. The latter is represented by sapphire, mica, and single-crystal gold [10,11,12]. The crystals exhibit van der Waals epitaxy, with specific orientations on the surface of such substrates and suitable growth conditions. This is beneficial for the controllable synthesis of MoS_2_, so this type of substrate has become a preferred choice. Among them, the growth of MoS_2_ on sapphire is widely studied. Firstly, as an industrial product, sapphire is easy to obtain and inexpensive. Secondly, C-plane sapphire possesses a six-fold symmetric structure, which matches the three-fold symmetry of MoS_2_, making epitaxial growth more likely to occur. Furthermore, sapphire substrates can be polished and annealed to form surfaces with low roughness, which is very beneficial for material growth and transfer. In addition, the lattice constant and in-plane expansion coefficient of sapphire are greater than those of MoS_2_, which ensures that the material is always stretched on the substrate surface and does not develop wrinkles. Therefore, studying the growth behavior of MoS_2_ on sapphire substrates is of great significance for the preparation and application development of MoS_2_.

The surface of sapphire can generate steps after annealing, which have a significant impact on the growth of MoS_2_. Li et al. and Wang et al. prepared wafer-scale MoS_2_ single crystals by designing the step direction of the substrate [13,14]. Subsequently, Zheng et al. discovered that the immature steps promote grain nucleation near the step edges and guide the unidirectional alignment regardless of the step direction [15]. Furthermore, Liu et al. found that steps with a height of about 1.26nm can induce the nucleation of double-layer MoS_2_ [16]. In addition, the theoretical calculations indicated that different step heights regulate the nucleation of MoS_2_ with different layer thicknesses, providing insights for the preparation of multi-layer materials [17].

Here, we utilized different annealing processes to obtain low-step (0.44–1.54 nm) sapphire substrates (LSSs) and high-step (1.98–3.30 nm) sapphire substrates (HSSs). Unidirectional MoS_2_ triangle grains and nanoribbons were achieved on LSS and HSS, respectively. Nanoribbons were formed with multiple grains, and the single-crystal nature of the nanoribbons was demonstrated through TEM and SHG measurements. Theoretical calculations conducted using CP2K suggest that the high steps promote the nucleation of monolayer MoS_2_ and prevent cross-step growth. This work sheds light on the growth of monolayer MoS_2_ and its potential in various applications.

## 2. Materials and Methods

### 2.1. Annealing of Sapphire

We chose c-plane sapphire (HeFei crystal Technical Material, Hefei, China) with a miscut angle of ~1° towards the A-axis (C/A) as the substrate. Before the growth, the substrates were annealed under atmospheric pressure to generate steps on the surface. Low-step (0.44–1.54 nm) sapphire substrates (LSS) were obtained through annealing at 1000 °C for 3 h and then cooling to room temperature at a rate of 10 °C/min, while high-step (1.98–3.30 nm) sapphire substrates (HSS) were produced through annealing at 1100 °C for 3 h and naturally cooling to room temperature.

### 2.2. Synthesis of MoS_2_

The growth of MoS_2_ was performed in a low-pressure (150 Pa) CVD system (Anhui BEQ Equipment Technology, Hefei, China). S powder (purchased from Alfa, 99.995%, 10 g) was loaded in the main tube and carried by 100 SCCM Ar, while MoO_3_ powder (Alfa, 99.95%, 20 mg) was loaded in the tiny tube and carried by 50 SCCM Ar. A piece of annealed sapphire was placed downstream as the substrate. The temperature of S, MoO_3,_ and the substrate was 200 °C, 580 °C, and 850 °C, respectively. The growth duration was maintained at 15 min, and then the furnace was naturally cooled to room temperature.

### 2.3. Sample Characterizations

The morphology of the MoS_2_ samples was obtained with an optical microscope (OM, BX51M, Olympus, Tokyo, Japan), a scanning electron microscope (SEM, Sigma HD, Zeiss, Oberkochen, Germany), and an atomic force microscope (AFM, ICON, Bruker, Billerica, MA, USA). Raman, photoluminescence (PL), and second-harmonic generation (SHG) were recorded on a confocal laser microscope system (Alpha 300RS+, WITec, Ulm, Germany). Selected-area electron diffraction (SAED) patterns and high-resolution TEM (HRTEM) images were obtained via a transmission electron microscope (TEM, F200S G2, Thermo Fisher, Waltham, MA, USA).

### 2.4. CP2K Calculation

All calculations were performed with periodic DFT using the Gaussian and Plane Wave (GPW) method implemented in CP2K’s Quickstep module (2023, version 2023.1; the CP2K developers group) [18,19]. The explorative studies of these structures were performed using the molecularly optimized basis set DZVP-MOLOPT-SR-GTH for each atom with a Goedecker–Teter–Hutter (GTH) pseudopotential [20,21,22]. The calculations were conducted using the generalized gradient approximation and the Perdew–Burke–Ernzerhof (PBE) functional [23] with D3 (BJ) correction [24]. The orbital transformation method was used to converge the self-consistent field (SCF) cycle with an electronic gradient tolerance value of 5 × 10^−6^ Hartree. An energy cutoff of 350 Ry was used throughout the calculations. Vacuum spacing larger than 15 Å was introduced to avoid artificial interaction between the periodic images along the z-direction. The input file was generated with Multiwfn software (2023, Multiwfn version 3.8(dev), Beijing Kein Research Center for Natural Sciences, Beijing, China) [25].

## 3. Results and Discussion

An LSS and HSS were used for the substrate to investigate the effect of step height on the growth of monolayer MoS_2_. Figure 1a,e illustrate the step height distribution of the LSS and the HSS, respectively. It is observed that, on the LSS, the step height ranges from 0.44 to 1.54 nm and is concentrated at 1.10 nm, whereas the steps on the HSS are quite steep, and its height distribution is more extensive (1.98−3.30 nm). An optical microscopy (OM) image of the MoS_2_ grains grown on the LSS (Figure 1b) shows a typical triangle shape with a uniform orientation. In contrast, the MoS_2_ grains grown on the HSS (Figure 1f) exhibit a nanoribbon shape with a uniform orientation. The detailed morphology of the two types of grains was characterized through AFM. As shown in Figure 1c,g, the thickness of the triangle and nanoribbon grains is 0.68 and 0.72 nm, respectively, indicating that both grains are monolayer. The triangle grains grow across the step, and one side of the grain is parallel to the edge of the step, which is consistent with the result of the literature [13]. However, the growth of the MoS_2_ nanoribbons ceases at the edge of the step, and the nanoribbons share the same direction with steps. As shown in Figure 1d,h, the Raman spectra of the MoS_2_ grown on different steps show two peaks, E_2g_ (~386 cm^−1^) and A_1g_ (~405 cm^−1^), with a peak position difference of ~19 cm^−1^. No noticeable differences in the peak position, full width at half maxima (FWHM), and intensity are observed, indicating that the step height does not affect the quality of MoS_2_. Moreover, the PL spectra of these two types of MoS_2_ are consistent in Appendix A, which confirms the above conclusion.

SEM was performed to investigate the evolution of the morphology of the nanoribbons over the deposition time. As shown in Figure 2a, scattered grains form on the substrate surface when the growth lasts for 5 min. If the deposition time increases to 10 min, these grains grow up, and the adjacent grains begin to splice. Meanwhile, grains stop growing at line 1 and line 2. The inserted AFM image in Figure 2b reveals that the growth stops at the edge of the steps. When the growth time reaches 15 min, the grains complete stitching and form a nanoribbon. Therefore, MoS_2_ nanoribbons are not grown from a single grain but are formed through the stitching of multiple grains.

To verify the orientation of the grains and the associated crystallinity of the MoS_2_ nanoribbons, the samples were characterized with TEM and SHG. Figure 3a shows the transferred nanoribbons, marked with two dashed lines. A HRTEM image of a nanoribbon (Figure 3b) reveals clear light/dark contrast, enabling the identification of Mo and S atoms. The crystal displays a near-perfect atomic lattice, indicative of high crystallinity. Furthermore, the high crystallinity of the triangle crystal is also revealed in Appendix A, which confirms that the step height does not influence the quality of MoS_2_.The SAED pattern of a nanoribbon is shown in Figure 3c, and the MoS_2_ grains exhibit one set of hexagonal diffraction spots. We compared the diffraction spot angles in 150 regions and found that the angles were consistent (Figure 3d), which reveals that the grains possess a highly aligned orientation. However, MoS_2_ crystals belong to space group p63/mmc, meaning that the diffraction spots of the 0° and 60° orientations are equivalent. We cannot recognize the differences between these two types crystals via SAED. Grains stitched by different orientations would produce grain boundaries [26,27]. When monolayer MoS_2_ crystals are stitched with a misorientation of 60°, a 4|4E-type grain boundary forms. The SHG signal is quenched at this type of grain boundary, making it detectable via intensity mapping [28]. The SHG mapping (Figure 3e) of a nanoribbon exhibits a uniform intensity distribution, indicating the absence of 4|4E-type grain boundaries. Therefore, all grains share the same orientation, and the stitched MoS_2_ nanoribbons are monocrystalline.

To better understand the different growth behaviors of the two surfaces, CP2K calculations were further conducted. Firstly, we calculated the difference in binding energy, whereas the binding energy of MoS_2_ on a flat surface is set as the energy reference. As shown in Figure 4a, the binding energy difference changes from –1.79 eV for a 1.54 nm-high step to –1.86 eV for a 1.98 nm-high step, suggesting that, energetically, the MoS_2_ domain prefers to nucleate at higher steps. Second, the “step-climbing” energy barriers were calculated to be 8.48 eV and 8.61 eV for a 1.54 nm-high step and a 1.98 nm-high step (Figure 4b), which indicates that climbing across a higher step is more difficult. Moreover, the calculation for a 0.66 nm-high step confirms this conclusion, as shown in Appendix A.

Based on the above-mentioned results, we proposed the following mechanism for the growth of MoS_2_ on the LSS and HSS. Firstly, due to the lower nucleation energy at the step edge and the design of the step direction, highly oriented crystals are formed at the step edge on both substrates, as shown in Figure 5a,d. With an increasing deposition time, the grains continue to grow. Since the energy required to cross the step on the surface of the LSS is relatively low, the sample exhibits cross-step growth (Figure 5b,c). However, the energy required for crossing the step is too high on the HSS. Hence, as shown in Figure 5e, the crystals nucleating at the same step edge cannot cross the step (i.e., grow within the step surface). Eventually, it forms nanoribbons in the same direction as the step (Figure 5f).

## 4. Conclusions

In summary, we demonstrated the effect of the step height of the sapphire substrate on the growth of monolayer MoS_2_. Triangular crystals with a consistent orientation were prepared on an LSS, while unidirectional nanoribbons were obtained on a HSS. The nanoribbon was formed by multiple grains and proved to be a single crystal. Step height did not influence the quality of the MoS_2_. The CP2K calculations revealed the promotion of nucleation and the inhibition of cross-step growth with larger step heights. The results shed light on the growth mechanism of monolayer MoS_2_ and could promote its applications in electrical, optical, and energy-related devices.

## Figures and Tables

**Figure 1 nanomaterials-13-03056-f001:**
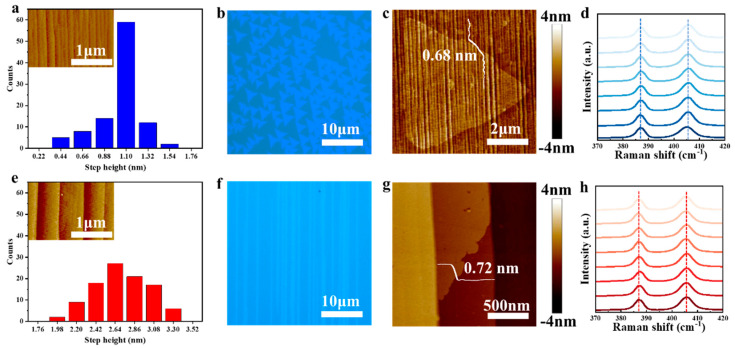
Step height distribution and the growth results of monolayer MoS_2_ on LSS and HSS. Statistics on the step height of LSS (**a**) and HSS (**e**). OM images of MoS_2_ on LSS (**b**) and HSS (**f**). AFM images of MoS_2_ grown on LSS (**c**) and HSS (**g**). Raman spectrum of MoS_2_ on LSS (**d**) and HSS (**h**).

**Figure 2 nanomaterials-13-03056-f002:**
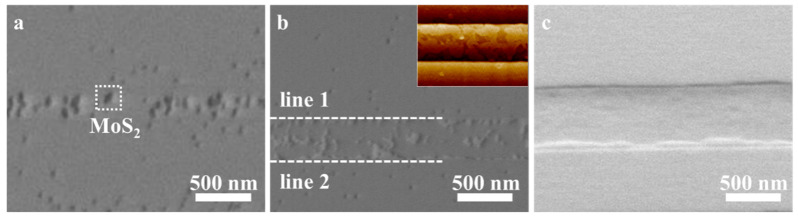
Evolution of morphology of MoS_2_ nanoribbons with growth time. SEM images for 5 min (**a**), 10 min (**b**) with an inset of the corresponding AFM image, and 15 min (**c**).

**Figure 3 nanomaterials-13-03056-f003:**
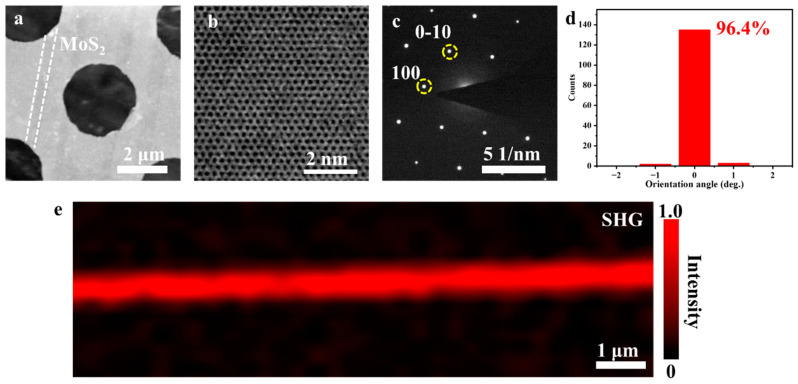
The crystallinity of MoS_2_ grains of nanoribbons. (**a**) Low-magnification TEM image of nanoribbons and corresponding high-resolution TEM image (**b**) of SAED pattern (**c**). The (0−10) crystal plane in (**c**) is defined as the x-direction, and thus the orientation angle in (**d**) is defined as the angle between the (0−10) crystal plane and the x-direction. (**e**) SHG mapping of a MoS_2_ nanoribbon.

**Figure 4 nanomaterials-13-03056-f004:**
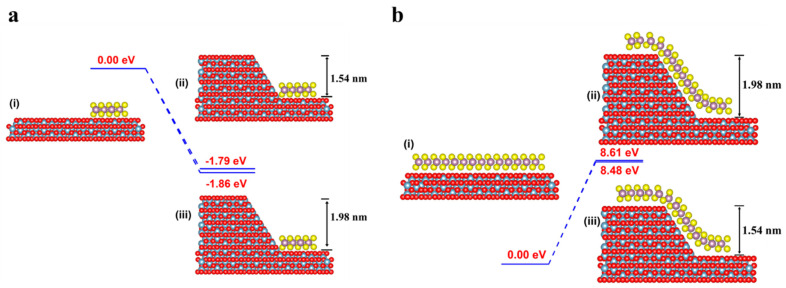
CP2K calculations of the growth. (**a**) Energy profiles of MoS_2_ on a flat surface (i) attached to a 1.54 nm-high step (ii) and a 1.98 nm-high step (iii). (**b**) Energy profiles of MoS_2_ on a flat surface (i) crossing a 1.54 nm-high step (ii) and a 1.98 nm-high step (iii), respectively. The binding energies of MoS_2_ on a flat surface are set as energy references.

**Figure 5 nanomaterials-13-03056-f005:**
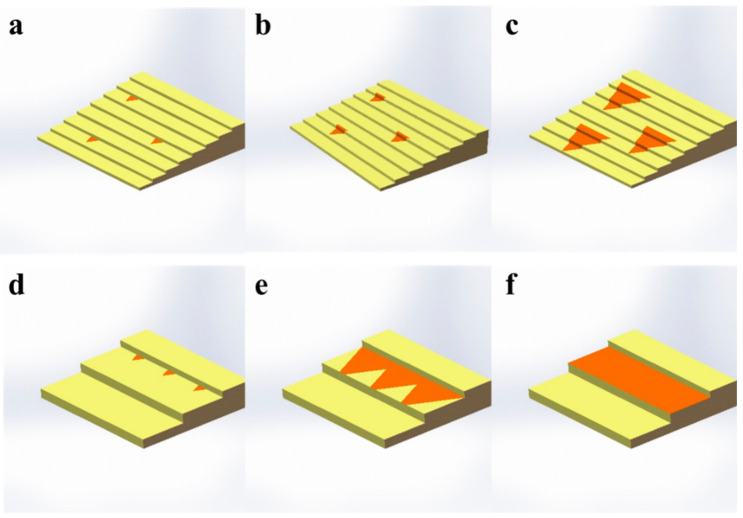
The schematic diagram for the growth of monolayer MoS_2_. Growth on LSS (**a**–**c**) and HSS (**d**–**f**).

## Data Availability

The data presented in this study are available from the corresponding author upon reasonable request.

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
