# Peer review of "Impact of Sapphire Step Height on the Growth of Monolayer Molybdenum Disulfide"

_nanomaterials, 2023, doi:10.3390/nano13233056_

Round 1

Reviewer 1 Report

Comments and Suggestions for Authors

The manuscript presented by Lu et al. discusses how the step height of the sapphire surface influences the morphology of MoS2 grown on it.

In Figure 3a, it appears that something is covering the hole in the carbon supporting film of the TEM grid. What could that be? Additionally, the caption for Figure 3 should read "Low-magnification" instead of "Low-resolution."

The authors used SHG mapping to conclude that the polycrystalline MoS2 nanoribbon is absent of grain boundaries. However, what is the resolution of the SHG technique for determining the twist angle of the MoS2 grains? It's worth considering a reference paper (Chem. Mater. 2018, 30, 403-411) which illustrates the atomic structure of the stitching grain boundary with a misorientation angle of less than 4 degrees. This likely applies to the present work on MoS2. The author should rephrase this sentence and provide proper citation.

In Figure 4, the authors propose atomic models of the MoS2 growth over the step edge of sapphire. The results suggest that the step edge with a slope is crucial for enabling MoS2 to climb over or step down the edges. Have the authors ever identified step edges of sapphire with a slope using AFM?

I recommend adding one more calculation for the low step substrate (step height < 1.54nm) to represent the formation energy of the triangular MoS2 domain grown on the condition with a slope and without a slope for comparison.

Overall, the paper effectively explains how the domain size of the MoS2 grown on sapphire is limited by the step height of the sapphire surface. It has the potential for publication in Nanomaterials once these comments are addressed.

Reviewer 2 Report

Comments and Suggestions for Authors

1)    The authors have shown the effect of their low-step sapphire (LSS) and high-step sapphire (HSS) influencing different type of molybdenum disulfide (MoS2) growth. Do the authors have any optical properties like PL of their grown MoS2?

2)    Since the authors had mentioned that their MoS2 nanoribbon is epitaxial and taken TEM imaging and SAED images, can the authors share some high-resolution TEM image of their MoS2 nanoribbons?

Reviewer 3 Report

Comments and Suggestions for Authors

In this manuscript, Jie Lu et al. demonstrated the impact of sapphire step height on the growth of monolayer MoS2. CP2K calculations show that MoS2 domain prefers to grow at higher steps while climbing across a higher step is more difficult. In my opinion, the authors should make some changes and clarify a few points in the manuscript before it can be accepted for publication. The details are as follows.

  1. As mentioned by the authors that “The results shed light on the growth mechanism of monolayer MoS2 and could promote its applications.” The authors should give at least one example.
  2. How will the step height affect the quality of the obtained monolayer MoS2 nanosheets? Raman spectra from different parts should be tested and compared.
  3. The authors should provide the high-quality HRTEM images of the obtained monolayer MoS2 nanosheets.
  4. The authors should check the format, spelling, and grammar carefully. There are some typos and grammar mistakes. For example, in line 137, “SEAD” should be changed to “SAED”; in the reference part, page numbers of some references should be added.
Comments on the Quality of English Language
  1. The authors should check the format, spelling, and grammar carefully. There are some typos and grammar mistakes. For example, in line 137, “SEAD” should be changed to “SAED”; in the reference part, page numbers of some references should be added.

Reviewer 4 Report

Comments and Suggestions for Authors

In this manuscript, the authors present a study of the impact of sapphire step height on the growth of monolayer MoS2 by chemical vapor deposition (CVD). The work claims that MoS2 exhibits a highly oriented triangular grain on a low step substrates, while nanoribbons with consistent orientation on a high step substrates are observed. Thus, the present very timely work provides an original investigation and a well-substantiated study of the physics behind the CVD growth of MoS2 on sapphire. Moreover, the authors employ characterization techniques (SEM, TEM) and DFT (CP2K) numerical simulation setup that both are chosen adequately. Consequently, the conceptualization of this work, as well as the results presented are both original and realistically applicable to a wide range of problems in the realm of CVD growth of 2D materials on different (not only sapphire) substrates. Also importantly, the discussion of the results not only looks credible but inspires future work in the field.

The figures, their captions and their corresponding discussion in the main text are easy to understand and they are logically organized too.

This work certainly represents a valuable contribution with possible wider impact to the field.

There are however some relatively minor concerns about textual details of this already very good work that should be addressed before the manuscript becoming suitable for publication, i.e., it can be considered for publication after a minor revision:

1: Abstract should provide clear and concise reference to the level of theory employed DFT/PBE/D3 (which is more important in the present case than the implementation – the  CP2K’s Quickstep module). Such clarity will make the work much more attractive to the audience.

2: More technical details about the crystallinity of MoS2 grains of nanoribbons as presented in Fig. 3 should be discussed in the main text.

3: Have the authors checked/tested/experimented with different exchange correlation functionals, other than PBE, when calculating the energy profiles of MoS2?

4: The authors miss to mention existing simulation work done for similar material systems by using DFT as well as ab initio molecular dynamics (where the role of substrates, capping layers etc. have been considered for the CVD growth of 2D materials) which then worked successfully as a guidance to experimental work, namely (CrystEngComm 23 (2021) 385-390; Physical Chemistry Chemical Physics 25 (2023) 829-837; Nat. Mater. 15 (2016) 1166). As it is currently, the introduction of the manuscript lacks a broad view.

Comments on the Quality of English Language

Spell-check and stylistic revision of the paper are necessary. Some long sentences, as well as misspellings, etc., are noticeable throughout the text.

Round 2

Reviewer 3 Report

Comments and Suggestions for Authors

The authors have addressed most of the comments and revised the manuscript properly. I have no more comments. Finally, I would like to recommend this manuscript to be accepted.